# Antifungal Resistance in Dermatophytes: Genetic Considerations, Clinical Presentations and Alternative Therapies

**DOI:** 10.3390/jof7110983

**Published:** 2021-11-18

**Authors:** Rosalie Sacheli, Marie-Pierre Hayette

**Affiliations:** Belgian National Reference Center, Department of Clinical Microbiology, University Hospital of Liege, 4000 Liège, Belgium; mphayette@chuliege.be

**Keywords:** dermatophytes, resistance, terbinafine, azoles

## Abstract

Numerous reports describe the emergence of resistance in dermatophytes, especially in *T. rubrum* and *T. mentagrophytes/indotineae* strains. We here present a review of the current status of resistance in dermatophytes worldwide. Resistance to terbinafine is mainly discussed, with different mutations found in the squalene epoxidase gene also considered. Resistance to azoles is also approached. Clinical presentations caused by resistant dermatophytes are presented, together with alternative therapies that help to better manage these kind of infections.

## 1. Introduction

Treatment resistant dermatophytoses caused by *Trichophyton rubrum (T. rubrum)* or *Trichophyton mentagrophytes (T. mentagrophytes)/Trichophyton indotineae* have recently emerged as a global public health issue. This phenomenon is spreading, and is particularly important in endemic areas such as India. In Europe and other industrialized countries, several cases have been reported. Most dermatophyte resistance concerns terbinafine (TERB) and is characterized by point mutations in the squalene epoxidase (SE) gene. This review will focus on the phenomenon of resistance to antifungals encountered in dermatophytes, with a special focus on terbinafine resistance and mutations in the SE gene. Resistance to azoles will also be considered. A clinical review of lesions caused by these resistant dermatophytes will be presented and alternative therapies will be discussed.

## 2. Focus on Terbinafine Resistance in Dermatophytes

### 2.1. First Reports of Terbinafine Resistant Associated Cases

The first case of terbinafine resistant *Trichophyton rubrum* was recorded in North America by Mukherjee et al. in 2003 [1]. In this work, six isolates of *T. rubrum* from the same patient were characterized to determine minimum inhibitory concentrations (MICs) of terbinafine from 4 µg/mL to >128 µg/mL. At this time, no molecular characterization of the strains was carried out, except random amplified polymorphic DNA (RAPD) analyses that did not reveal any differences between terbinafine resistant isolates and susceptible ones. The paper also reveals that terbinafine resistant isolates show normal susceptibilities to antifungals such as itraconazole, fluconazole and griseofulvin, but they are crossresistant to several other known squalene epoxidase inhibitors, such as naftifine, butenafine, tolnaftate and tolciclate, suggesting a target specific mechanism of resistance to allylamines [1].

In 2004, Favre et al. investigated the biochemical basis for resistance in the six sequential isolates of *T. rubrum* from the same patient reported by Mukherjee et al. They showed that resistant strains had normal ergosterol biosynthesis but a reduced accumulation of radiolabelled squalene, suggesting reduced squalene epoxidase activity. They showed that squalene epoxidase from resistant strains was three orders of magnitude less sensitive to terbinafine than the normal enzyme, signalling that the resistance to terbinafine found in the six *T. rubrum* isolates is due to alterations in the squalene epoxidase (SE) gene. At this time, the authors suggested that amino acid substitutions are likely be fully responsible for terbinafine resistance in *T. rubrum* [2]. Figure 1 describes the mechanism of terbinafine resistance (in A) but also (in B) the intervention of ABC transporters in azoles resistance, discussed in point 3.

The suggestion that amino acid substitutions are likely be fully responsible for terbinafine resistance in *T. rubrum* was confirmed in 2005 by Osborne et al., who characterized an amino acid substitution in the squalene epoxidase of *T. rubrum* for the first time. They highlighted the presence of an intron in the gene and an open reading frame encoding a protein of 489 residues. In terbinafine resistant strains serially isolated from the same patient (from the previous study of Mukherjee et al., 2003), they found the amino acid substitution L393F. Introduction of the corresponding substitution to the SE of *Candida albicans* and the transfer and expression of this gene in *Saccharomyces cerevisiae*, resulted in a profile of resistance to terbinafine in transformants, confirming that the single amino acid substitution L393F is responsible for the reduced affinity of the antifungal to squalene epoxidase [3].

Later, in 2006, Osborne characterized another amino acid substitution, F397L, caused by a missense mutation in the squalene epoxidase gene. This mutation was found in a strain of *T. rubrum* from Switzerland. The strain exhibited an MIC for terbinafine of 64 µg/mL and showed crossresistance against other SE inhibitors. However, no modification of the susceptibility to fluconazole and griseofulvin was observed [4].

In 2015, Alipour et al. used random amplified polymorphic DNA (RAPD) to classify terbinafine resistant strains of *T. mentagrophytes*, compared to susceptible ones. They observed a good separation between resistant and susceptible strains but some resistant strains were still grouped with susceptible strains, showing the imprecision of this method for characterizing terbinafine resistant strains. The use of molecular sequencing of the SE gene thus remains the best way to characterize strains that are resistant to terbinafine [5].

### 2.2. The Indian Story

Since 2018, alarming information has come from India regarding high terbinafine resistance in isolates initially named “*Trichophyton interdigitale”*. The correct identity of the etiological agent causing Indian dermatophytosis epidemics is a much debated topic [6,7]. In order to define species limits, a taxonomic study was conducted by Tang et al. in 2021, combining molecular, morphological and physiological characteristics to classify the strains responsible for the Indian epidemics. The conclusion of the study is that the Indian strain can be distinguished from the *T. interdigitale/mentagrophytes* group based on a “high mobility group” (HMG) sequence, therefore, the name *T. indotineae*, previously suggested by Kano et al. in 2020 [8], was retained for the Indian clone (formerly *T. mentagrophytes* type VIII) [9]. However, based on the understanding of current established dermatophyte taxonomy, in 2021 Verma et al. contested this classification as a new species, as they consider that *T. mentagrophytes* type VIII is only one variety within the cluster of a large number of genotypes of the *T. mentagrophytes/T. interdigitale* complex. Therefore, they consider that it is not appropriate to attempt to assign this single genotype VIII of *T. mentagrophytes* to a distinct new species [10]. In this review, to avoid confusion, we will use the name used on the original paper by the authors.

In 2018, Singh et al. reported a resistance rate of 32% among *T. interdigitale* strains in India. All strains were characterized by high MIC values for terbinafine, comprised between 4 and 32 µg/mL. All isolates harboured one of the single point mutations F397L (in twelve strains) or L393F (in eight strains). This was the first report of the mutation L393F in *T. interdigitale*, this mutation has previously been highlighted in *T. rubrum*, as described above in this review. All the resistant strains were isolated from extended tinea corporis/cruris, so this was also the first report of *T. interdigitale* resistant strains other than those giving rise to difficult to treat onychomycoses [11].

In addition, in India, in 2018, Rudramurthy et al. published a retrospective study conducted among patients with dermatophytosis in 2014. A total of 127 *Trichophyton* isolates were submitted to antifungal susceptibility testing against twelve antifungals, including terbinafine. Among the fifteen *T. interdigitale* and five *T. rubrum* strains showing high MIC values for terbinafine, the substitution F397L was found in four *T. interdigitale* and two *T. rubrum* isolates. Again, here, the incidence of strains resistant to terbinafine is high (15.7%), even if lower than in the study of Singh et al. This study also shows that a profile of resistance to terbinafine with MICs > 2 µg/mL is not always correlated with a mutation of the SE gene, as in the study fourteen strains out of twenty with high MIC values for terbinafine did not present any SE mutation [12].

At the end of 2018, another report came from India from Khurana et al., correlating laboratory data with clinical responses in thirty tinea corporis/cruris cases with complete follow-up data. All cases were due to *T. interdigitale.* Antifungal susceptibility testing was performed and SE gene analysis was carried out on some strains (18/30 cases). All strains shared MIC values for terbinafine >=0.5 µg/mL (0.5 -> 32 µg/mL). The amino acid substitution F397L was found in ten strains, while the L393F substitution was present in three strains. Five strains did not harbour any substitution in SE. In this study, a correlation was made between a high exposure to terbinafine, an MIC value >8 µg/mL and the presence of an SE mutation [13].

In 2019, Burmester et al. described a clinical case of extended tinea cruris due to a *T. mentagrophytes* strain presenting the amino acid substitution F397L in SE in an Indian male [14]. In 2019, Singh et al. defined a unique multidrug resistant clonal *Trichophyton* population distinct from the *T. mentagrophytes/interdigitale* complex that was causing an alarming dermatophytosis outbreak in India. Genome analysis of the *Trichophyton* species causing severe and extended dermatophytosis in North India confirmed that the strains belong to a unique diverging clade related to *T. interdigitale/mentagrophytes*. The clonal origin was confirmed to show forty-two single point mutations (SNPs) compared to *T. interdigitale/mentagrophytes*. Among this clade, high rates of resistance were observed for terbinafine (MIC range 4–32 µg/mL) but also for fluconazole (MIC range 32–64 µg/mL) and griseofulvin (geometric mean MIC ≥ 4 µg/mL). Amino acid substitutions L393F and F397L were found in the SE protein of all the tested terbinafine resistant isolates. Therefore, this study identified a new population among *Trichophyton sp*. with high rates of in vitro antifungal resistance. This population seems to be responsible for the ongoing outbreak of dermatophytosis in India [15].

In 2020, Ebert et al. conducted an epidemiological study across India (in eight different locations), including 402 patients with clinically suspected dermatophytosis. Among the isolates, 314 (78%) were identified as *T. mentagrophytes* type VIII, eighteen (5%) were from the *T. interdigitale/mentagrophytes* group and nineteen (5%) were *T. rubrum*. This repartition is quite different than that observed in European dermatophytosis. Among these isolates, 71% were resistant to terbinafine, the amino substitution F397L was found in 91% cases and two novel substitutions were revealed: S395P and S443P. The substitution A448T was found in both terbinafine susceptible and resistant strains but was also associated with increased MICs for voriconazole and itraconazole. They observed that triazoles resistance was more frequent in terbinafine susceptible strains than in resistant ones. This study shows a troubling rate of terbinafine resistance in India, which has continuously increased since the reports from 2018 [16].

Later in 2020, Burmester et al. characterized strains from a large Indian collection of *T. mentagrophytes* showing mutations within SE. The highest MIC values for terbinafine were found for L393F and F397L mutants. Amino acid substitution Q408L also conferred terbinafine resistance. This substitution was previously described in a paper by Hsieh in 2019 [17]. Burmester et al. also observed that A448T single mutants were terbinafine sensitive, but about 50% of the isolates showed increased fluconazole resistance. Double mutants F397L/A448T demonstrated higher crossresistance to terbinafine and fluconazole (MICs twice or thrice as high compared with wild type strains, and twice compared to single mutant F397L) demonstrating a selective advantage of the combination of both mutations. They suggest, in the work, that the A448T substitution may protect against the fungicidal effects of terbinafine [18].

In 2020, Shankarnarayan et al. evaluated the rapid detection of terbinafine resistance by amplified refractory mutation system polymerase chain reaction (ARMS PCR). In the study, they showed that this was a simple and reliable method to detect terbinafine resistant *Trichophyton* isolates. They tested 214 dermatophyte strains and, among these, high MICs > 2 µg/mL were detected in fifteen (15.4%) *T. mentagrophytes* isolates. All had an amino acid substitution in position 397 of SE [19].

At the end of 2020, Gaurav et al. described six cases of recalcitrant dermatophytosis in patients attending the dermatology unit of an Indian tertiary care hospital. The etiological agent was identified as *T. mentagrophytes*. MIC values observed for terbinafine were between 0.125 and 8 µg/mL. Three isolates presented the F397L substitution, one of them bore the double substitution F397L/Y394N never described before and this double mutation was also associated with high MIC values for fluconazole and itraconazole. Two of the resistant strains had no SE mutations [20]. All these reports highlight the high prevalence of terbinafine resistance in India. Due to poor sanitary conditions and little access to hospitals/laboratories, the situation in this country is certainly underestimated. We can also notice the high infectiousness of the *T. indotineae* strain, despite its zoophilic profile, which is exacerbated by the poor sanitary conditions.

### 2.3. Emerging Reports from Europe

The need for antifungal testing among dermatophytes was reinforced by the above alarming reports from India. Several cases have also emerged in Europe, although these are still sporadic and often linked to travel to endemic regions.

In 2017, observing high rates of terbinafine treatment failure in patients suffering from onychomycoses in Switzerland, Yamada et al. collected *T. rubrum* and *T. interdigitale* strains over a three-year period. A total of 2056 strains were screened by an agarose dilution method where 0.2 µg/mL of terbinafine is added to Sabouraud agar. Among them, seventeen strains (1%, sixteen *T. rubrum* and one *T. interdigitale*) were found to harbour SE gene alleles bearing point mutations at different positions. Different amino acid substitutions were described among resistant strains at positions Leu^393^, Phe^397^, Phe^415^ and His^440^. Among strains bearing a substitution, all exhibited a MIC for terbinafine > 0.1 µg/mL (0.1 -> 12.8 µg/mL). Specifically, the amino acid substitutions found in *T. rubrum* were as follows: L393F, L393S, F397L, F397I, F397V, F415I, F415S, F415V and H440Y. Therefore, seven new mutations of the SE gene were highlighted by this study. One strain of *T. interdigitale* also exhibited the F397L substitution [21]. This work encouraged scientists to be concerned by the terbinafine susceptibility of dermatophytes and implement in vitro testing.

In 2017, Wingfield Digby et al. reported a case of Darier disease complicated by terbinafine resistant *T. rubrum* showing a MIC > 4 µg/mL in a 62 year old man in Denmark. In 2018, again in Denmark, Schosler et al. described a case of recurrent onychomycosis due to *T. rubrum* in a nine year old boy. The strain presented an MIC for terbinafine of 4 µg/mL. No molecular investigations were described in either case, therefore, any mutation in the SE gene of *T. rubrum* is unknown [22,23].

In 2019 Hsieh et al. presented a case of disseminated tinea corporis in a couple in Switzerland due to terbinafine resistant *T. mentagrophytes*. The couple had visited their son in India. Resistance to terbinafine was established by first testing the ability of the isolated strain to grow on Sabouraud agar containing 0.2 µg/mL of terbinafine, as described by Yamada et al. [21]. The report highlighted a new mutation in the SE gene leading to a Q408L substitution in the protein [17].

Reports have been published of the spread of resistant strains outside of India. In 2019, Saunte et al. reported terbinafine treatment failure among Danish patients. Twelve *T. rubrum* and two *T. interdigitale* specimens were characterized to have high MICs for terbinafine between 2013 and 2018. Antifungal susceptibility testing was performed following the EUCAST E.Def 9.3.1 method. The SE gene was also sequenced in the study, twelve *T. rubrum* and two *T. interdigitale* presented an amino acid substitution in SE. Well known and novel SE amino acid substitutions were observed in this study, and were as follows: in *T. interdigitale*: F393L, L393F; in *T rubrum*: F393L, L393F L393S, F415S, and the newly described double substitutions H440Y/F484Y and I121M/V237I [24].

In 2019, in Germany, a case of extended tinea corporis was described by Suss et al. in a 6 month old baby from Bahrain, due to *T. mentagrophytes* of the Indian genotype. This case was molecularly characterized by the SE substitution F397L [25]. In addition, in Germany, in 2019 Burmester et al. described a case of extended tinea cruris in an Indian patient due to *T. mentagrophytes/interdigitale.* Internal transcribed spacer (ITS) PCR and squalene epoxidase sequencing defined the etiological strain as the genotype VIII *T. mentagrophytes* with a single point mutation at position 397 of the SE amino acid sequence [14].

In 2020, Nenoff et al. warned about the risk of spread of *T. mentagrophytes* type VIII by travel and migration in Europe and around the world. He described twenty-nine cases of *T. mentagrophytes* type VIII classified thanks to ITS and TEF1-alpha sequencing. These cases were from 2016 to 2020, all from German residents with or without contact with endemic countries such as India, Pakistan or Bangladesh. Among these *T. mentagrophytes,* they observed a MIC > 0.2 µg/mL for terbinafine in 13/29 (45%) strains. SE sequencing showed F397L substitution in ten strains, two shared a double substitution F397L/A448T and one strain had the L393F substitution. Among susceptible strains, they also found amino acid substitutions such as A448T, and one with A448T and V444I, newly described. Three strains also shared resistance to both itraconazole and voriconazole [26].

In 2020, Lagowski et al., a Polish team, described terbinafine resistance in *T. mentagrophytes* isolated from humans and also foxes (no mentioned contact with each other). The MICs obtained for terbinafine were between 16 and 32 µg/mL and all shared the L393F mutation. Other antifungals, including azoles, were tested without any elevated MICs reported. This work suggests that the terbinafine resistance phenomenon might not only be acquired by drug pressures, but can be intrinsic, as terbinafine resistant strains were isolated from asymptomatic animals [27].

In 2020, the Belgian National Reference laboratory reported the first Belgian case of terbinafine resistant *T. mentagrophytes*, isolated during a national survey of tinea capitis. The patient presented with tinea cruris associated with tinea capitis. No travel history could be associated with this case. The strain showed an MIC for terbinafine of 4 µg/mL and presented the mutation F397L [28].

In 2021, Siopi et al. conducted a study covering dermatophytoses from the last ten years in Greece. They studied in vitro susceptibility patterns with the newly described EUCAST E.Def 11.0 reference method for dermatophytes. The study was conducted on 112 dermatophytes, 70 *T. rubrum*, 24 *T. mentagrophytes*, 12 *T. interdigitale* and 5 *T. tonsurans*. No resistance to terbinafine was reported in *T.rubrum*, *T.interdigitale* or *T. tonsurans* species, while 9/24 strains resistant to terbinafine were found among *T. mentagrophytes*, with MICs comprised between 0.25–8 µg/mL. ITS sequencing defined these nine isolates as *T. mentagrophytes* type VIII, five of them bearing the F397L substitution, and the L393F substitution for the other four strains. No elevated MICs were reported for azoles or amorolfine in the study [29]. These reports from Europe show that terbinafine resistance is not only a problem for India, and even if the situation is not yet alarming, resistant strains are circulating in European countries and can rapidly spread and become a major public health concern.

### 2.4. Emerging Reports from Asia

In Asia, reporting of resistant strains has also increased in recent years. In 2018, Suzuki et al. described the first case of *T. rubrum* with low susceptibility to terbinafine in Japan. The strain exhibited a MIC > 128 µg/mL to terbinafine and the molecular characterization of the SE gene highlighted the L393F mutation. The strain exhibited normal susceptibility to itraconazole [30].

During 2018, Salehi et al. reported cases of terbinafine resistance in Iran among two species of dermatophytes *T. rubrum* and *T. tonsurans*, confirmed by the molecular observation of L393F substitution in the SE protein. Ninety-nine strains responsible for dermatophytoses were tested in the study for terbinafine resistance. Among them, two *T. rubrum*, two *T. tonsurans* and one *E. floccosum* strain showed reduced terbinafine susceptibility. However, among the five strains which showed reduced susceptibility to terbinafine, only two isolates (one *T. rubrum* and one *T. tonsurans*) showing an MIC for terbinafine > 32 µg/mL had the amino acid substitution L393F [31].

In 2019, Hiruma et al. determined the MICs of twenty-four strains of *T. interdigitale* against terbinafine and itraconazole, while no high MICs were observed for itraconazole, 1/24 strains tested showed an MIC value for terbinafine of 2 µg/mL. However, SE sequencing did not show any mutation in SE [32]. In 2019, Noguchi et al. described a case of tinea unguium in Japan caused by terbinafine resistant *Trichophyton rubrum*. The strain had an MIC for terbinafine ≥8 µg/mL and had a nucleotide substitution within the SE gene, leading to F397L substitution in the *T. rubrum* SE protein [33]. In a letter to the editor in 2020, Kakurai et al. described a case of extended tinea corporis due to a *T. interdigitale* strain in Japan that had an MIC for terbinafine of 32 µg/mL and bore an amino substitution F397L in SE [34]. In addition, in 2020, a Japanese team reported a case of extensive dermatophytosis due to *T. interdigitale* from the Indian genotype. The strain presented an MIC of 32 µg/mL for terbinafine and the substitution F397L was highlighted by SE sequencing [35]. Later in 2020, Hiruma et al. conducted an epidemiological study on terbinafine-resistant dermatophytes isolated from Japanese patients. Antifungal susceptibility testing was performed among *T. rubrum* and *T. interdigitale* strains, and SE sequencing was also performed. In the study, 210 isolates were characterized by the agar diffusion method described by Yamada et al. in 2017. Among the sixteen *T. rubrum* and one *T. interdigitale* that grew with a terbinafine concentration of 0.2 µg/mL, five strains (2.4%, all *T. rubrum*) showed MIC values for terbinafine > 32 µg/mL but remained susceptible to azoles. These five strains harboured an L393F substitution in SE. This epidemiological study shows the wide spread of terbinafine resistant dermatophytes around the world during recent years [36].

In 2020, Taghipour et al. presented an epidemiological study on 141 clinical isolates from different provinces of Iran isolated between 2016 and 2018. Among these strains, ninety six were *T. interdigitale* and forty five *T. mentagrophytes*. The majority of *T. interdigitale* specimens were isolated from tinea pedis, while the *T. mentagrophytes* were from tinea corporis cases. Among these, five *T. mentagrophytes* presented MICs for terbinafine > 32 µg/mL. All these five strains presented missense mutations in SE. The double substitutions L393S/A448T and F397L/A448T were highlighted in resistant strains. The single substitution A448T was also found in susceptible strains, as described in previous studies [37].

In 2020, Kano et al. suggested the new name “*Trichophyton indotineae* sp. nov.” for the Indian *T. mentagrophytes* type VIII clone. Based on ITS similarities, they could demonstrate that this species was slightly different from *T. interdigitale/mentagrophytes.* In this work, they analysed two strains isolated from Japan in 2019. The patients were regularly travelling to Nepal/India. The strains isolated were 100% identical to referenced terbinafine resistant Indian strains and showed three single polymorphisms, compared to the *T. interdigitale* reference strain. The F397L substitution was characterized in both strains [8]. The classification of *T. indotineae*, separate from *T. interdigitale/T. mentagrophytes*, was reinforced by the 2021 report by Tang et al. based on the HMG gene [9].

In 2021, a familial infection due to *T. mentagrophytes* type VIII was described in Iran by Fattahi et al. This concerned four people from the same family. The isolated strains showed huge MIC values for terbinafine ≥8 µg/mL but also for fluconazole (≥16 µg/mL). The amino acid substitution F397L was observed by SE sequencing [38]. In Iran, Firooz et al. conducted a study in 2021 with seventy one fungal isolates including dermatophytes (five *T. rubrum* and seven *T. mentagrophytes*). The authors do not mention how they chose the studied strain, but among the seven *T. mentagrophytes*, one (14%) was resistant to terbinafine with MICs ≥ 8 µg/mL. The strain presented the amino acid substitution F397L in SE [39].

A multicentric study was conducted and reported by Kong et al. in 2021. This concerned 135 isolates from India, China, Australia, Germany and the Netherlands. Thirty-five strains were identified as *T. mentagrophytes*, sixty four were *T. indotineae* (formerly *T. mentagrophytes* type VIII Indian clone) and thirty six were *T. interdigitale*. Among *T. indotineae,* 53% were resistant to terbinafine with MICs >16 µg/mL. All had the amino acid substitution F397L. Two isolates presented an MIC value of 0.5 µg/mL and the amino acid substitution F415V and L393S on SE. Amino acid substitutions K276N and L419F were also found in susceptible strains. Again, the double substitution F397L/A448T was associated with higher MIC values for triazoles, in addition to MICs > 16 µg/mL for terbinafine [40]. Table 1 is a summary of all the mutations found in SE all around the world and the incidence of resistance in different studies, when known. In addition, Figure 2 shows, on a world map, all mutations found in SE in different countries.

### 2.5. Reports from Other Continents

Case reports of terbinafine resistance were found in the USA, such as the report by Gu et al. in 2020 describing patients with extensive *T. rubrum* tinea corporis that persisted despite prolonged treatment with systemic and topical agents, including oral terbinafine [42]. However, except for the strains isolated by Mukherjee et al. in 2003 and molecularly characterized by Osborne et al. in 2005, no cases associated with SE amino acid substitutions have been described, to our knowledge. Reports from Africa or Oceania are still nonexistent, to our knowledge. However, strains from Australia were included in the multicentric study conducted by Kong et al. and described in Section 2.4 [40].

### 2.6. ABC Transporters in Terbinafine Resistance

Although the role of ATP binding cassette (ABC) transporters (e.g., TruMDR2) in terbinafine resistance in *T. rubrum* was mentioned by Fachin et al. in 2006 [43], the 2021 paper by Kano showed that no differences in PDR1, MDR2 and MDR4 transcript levels were found in *T. indotineae* when comparing terbinafine resistant strains to susceptible ones, indicating that ATP dependent efflux pumps do not seem to confer terbinafine resistance in *T. indotineae* [41].

### 2.7. Clinical Manifestations of Terbinafine Resistant Dermatophytes

Several cases of tinea have been reported, linked to dermatophytes sharing a profile of resistance to terbinafine. Here we summarize the main clinical presentations that are caused by these resistant dermatophytes.

#### 2.7.1. *T. rubrum* Clinical Manifestations

##### Onychomycoses

The first case of drug resistance causing recurrent onychomycosis due to *T. rubrum* was reported by Mukherjee et al. in 2006, but the authors did not describe the lesions and their clinical presentation [1]. In 2019, in a letter to the editor, Noguchi et al. reported a case of tinea unguium caused by terbinafine resistant *Trichophyton rubrum* in Japan. This concerned a 71 year old healthy female that presented whitis h discoloration and hyperkeratosis of the left index and little fingernails. A 10 month oral terbinafine treatment was unsuccessful. Fosravuconazole was then initiated and symptoms improved after three months. The isolated dermatophyte was *T. rubrum* with a MIC value for terbinafine >8 µg/mL and a mutation in SE (as described in Section 2.4). The authors suggest that fosravuconazole could be a promising alternative for treating tinea unguium in cases of terbinafine resistance [33].

In 2021, Appelt et al. described a case of resistant onychomycosis and tinea palmaris due to *T. rubrum* in Germany. The 49 year old patient had a 25-year history of onychomycosis. The clinical presentation was a yellowish discoloration of all toenails (left foot) with onychodystrophy and longitudinal grooves in the distal area of the nail plate. Palmar hyperkeratotic lesions were also present with fine lamellar scaling. SUBA-itraconazole was administered at 50 mg twice a day for one week, then once a week 2 × 50 mg SUBA-itraconazole. This was completed with a topical therapy of ciclopirox nail polish [44].

##### Skin Dermatophytosis

In 2017, Wingfield Digby et al. described a case of Darier disease complicated by terbinafine resistant *Trichophyton rubrum*. This concerned a 62 year old man who had experienced skin problems since the age of twelve. He presented tinea corporis with keratotic follicular papules and several well defined erythematous annular scaly plaques on his trunk and extremities. Cultures of skin scrapings were positive for *T. rubrum*. The patient was treated with oral terbinafine for several months. Due to the nonresolution of lesions, antifungal susceptibility testing was performed on the *T. rubrum* strain showing a MIC > 4 µg/mL for terbinafine. The MIC for itraconazole was low so the treatment was switched to oral itraconazole (100 mg/twice daily) leading to a clear improvement of the lesions [22].

In 2018, Schosler et al. reported a case of recurrent terbinafine resistant *T. rubrum* infection in a child with congenital ichtyosis. The patient developed itchy scaly skin between the toes. *T. rubrum* was identified as the etiological agent. Lesions recovered after two weeks of terbinafine treatment but clinical signs of dermatophytosis reappeared, showing a pruritic scaly erythematous rash on the abdomen, groin and legs. The treatment with topical terbinafine was supplemented with oral terbinafine (125 mg/d) for four weeks. This treatment was successful, but nine months later the rash reappeared again, caused by *T. rubrum*. Antifungal susceptibility testing was performed and the MIC of terbinafine was determined to be 4 µg/mL. Lower MICs were observed for azoles, so the treatment was changed to systemic itraconazole (100 mg/d) for three weeks, leading to a complete resolution of the rash [23].

Few cases of terbinafine resistance have been reported in the USA. In 2020, Gu et al. reported a case of extended dermatophytosis in a 45 year old patient. The first symptoms appeared twenty years before his presentation. The disease was first restricted to the toenails but extended to his feet, thighs, groin and buttocks. Diffuse annular erythematous plaques were seen with peripheral scale on his abdomen, lower back and thighs. Treatment with oral terbinafine for three months was unsuccessful and antifungal determination showed in vitro resistance to terbinafine so the treatment was switched to griseofulvin ultramicrosized, 500 mg twice daily with poor improvement. Itraconazole 200 mg daily for three months was finally administered [42].

In 2021, Appelt et al. described three cases of extended tinea corporis due to *T. rubrum* in Germany. The first case was a 62 year old woman with tinea corporis and faciei. A large area of polycyclic, erythematous, marginal plaques with fine lamellar scaling on the right flank and abdomen were observed, as well as on the cheek and neck. Disseminated scratching excoriations on the entire integument were also described. As an alternative therapy, this patient received itraconazole 100 mg once a day for four weeks, then once a week. The second case was a 44 year old man with tinea corporis, cruris and glutealis due to *T. rubrum.* Dry, scaly, erythematosquamous, irregularly delimited plaques were observed with accentuated edges asymmetrically on the right groin, right thigh, pubis and right gluteal. The patient was treated with itraconazole 200 mg every two weeks with bifonazole and ciclopirox cream once a day. The last case concerned a 49 year old man with tinea corporis. The clinical presentation was erythematosquamous plaques with fine scaling on the entire integument with an emphasis on the back and buttocks as well as both lower legs. This patient received SUBA-itraconazole 50 mg twice a day for two weeks, then once a week twice 50 mg for a further ten weeks [44].

#### 2.7.2. *T. mentagrophytes/interdigitale* Clinical Manifestations

In 2017, Verma et al. described a veritable epidemic of steroid modified tinea in India. They described a larger sized and greater number of lesions in different locations in individual patients, tinea cruris and corporis being more common. They discussed an increasing number of lesions with multiple concentric circles. Lesions have also been described as “tinea pseudoimbricata”. There was also mention of arciform lesions, an increasing number of multiple annular lesions of various sizes showing confluence, the dumbbell-shaped tinea formed by the juxtaposition of two large annular lesions with eczematized centres, and at times a curious clustering of multiple small annular lesions with active erythematous pustular borders. They mention antifungal resistance as the most important cause of treatment failure for the described dermatophytosis but did not isolate the dermatophyte or perform antifungal susceptibility testing [45].

In 2019, Shenoy et al. described difficult to treat dermatophytoses that shared atypical presentations, with extension of the disease into the scalp and face commonly reported. Typical tinea presents with clinical appearances ranging from eczematous, psoriasis-like, pustular lesions, pseudoimbricata (concentric rings), and rosacea-like lesions which are resilient to treatment. There is no mention of resistant dermatophytes in these descriptions due to a lack of laboratory diagnosis, most infections being diagnosed based on the clinical manifestations [46].

A large study including relapses and new cases of dermatophytosis was performed in India in 2014 and described in 2018 by Rudramurthy et al. Different clinical presentations were reported, such as tinea corporis, faciei, cruris, pedis and capitis. Different species of dermatophytes were isolated in 133 patients (68%), including, mainly, *T. interdigitale* (68%) and *T. rubrum* (26.3%) but also *Nannizia gypsea*, *M. canis* and *T. tonsurans*. Spreading lesions were detected in 159/197 (81%) of the patients. Sixty percent of the patients were classified to have recurrent dermatophytosis. The majority of the patients (63%) attempted self treatment before seeking medical attention. The study reported high MICs for terbinafine in 17.6% of *T. interdigitale* (2–32 µg/mL) strains and 14.3% of *T. rubrum* (2–16 µg/mL). The majority of the terbinafine resistant isolates also showed elevated MICs for naftitine [12].

In 2018, Pathania et al. presented a prospective study of the epidemiological and clinical patterns of recurrent dermatophytosis at a tertiary care hospital in India. Among 1600 patients with dermatophytes infections, 150 (9.3%) presented a recurrent form of the disease. An itching or burning sensation was frequently reported and multiple body sites were often concerned (tinea corporis, cruris, faciei, pedis and mannum). Extensive lesions (>10% of body surface area) were described in 14.7% of the patients. The clinical presentation was often annular lesions. Papulosquamous, eczematous, pustular, pseudo imbricata, lichemoid, pityriasis rosea like and bullous lesions were also reported even less frequently. The main etiological agents isolated from these lesions were *T. mentagrophytes* followed by *T. rubrum* and *T. interdigitale*. Antifungal susceptibility testing showed high MIC values for terbinafine for several strains [47].

In 2019, Hsieh et al. reported disseminated tinea corporis in a couple who had travelled in India, a 60 year old man and 51 year old woman. The clinical examination showed large annular plaques on the legs, arms and inguinal folds, without lymphadenopathy. The couple was treated with oral terbinafine 250 mg/day combined with topical terbinafine but the two-week treatment failed to improve the lesions. A culture of skin scrapings was positive for *T. mentagrophytes* and antifungal susceptibility testing established terbinafine resistance. Itraconazole was then successfully introduced 200 mg daily for 2–3 weeks with topical eberconazole [17].

In addition, in 2019, Burmester et al. described a case of dermatophytosis in a 26 year old male Indian patient. He presented livid extensive centrally scaly plaques featuring an erythematous edge and 1 mm pustules on the lower abdomen as well as bilateral inguinal reaching the gluteal region. *T. mentagrophytes* was the etiological agent of the lesion. SE substitution was found in *T. mentagrophytes* as previously described in Section 2.2. The therapy was changed from terbinafine to itraconazole combined with the application of ciclopiroxolamine cream [14].

In 2019 in Germany, Suss et al. described extensive tinea corporis due to a terbinafine-resistant *T. mentagrophytes* type VIII in a 6 month old infant from Bahrain. The baby presented extensive dermatophytosis of the back, buttocks, chest and groin. Treatment with terbinafine for two months failed. The lesions were successfully treated with topical miconazole and later by ciclopiroxolamine [25].

In 2020, Kimura et al. reported the arrival of terbinafine resistant *T. mentagrophytes/interdigiale* of the Indian genotype in Japan, causing extensive dermatophytosis. The patient concerned was a 27 year old healthy Nepalese woman. She travelled to Nepal ten months previously and stayed near to the border with India. She presented an extensive rash with multiple annular well demarcated erythemas and pigmentation on the cheeks, trunk, groin and axillae. *T. mentagrophytes/interdigitale* was isolated in culture and antifungal testing showed a MIC > 32 µg/mL for terbinafine. Consequently, ravuconazole (100 mg/day) was administered per os for four weeks but without any success. Finally, symptoms were cured by a combination of oral itraconazole (100 mg/day) and topical lulicoazole [35].

In addition, in Japan, in 2020, Kakurai et al. described extended dermatophytosis caused by *T. interdigitale* in a 47 year old healthy male who moved from India to Japan. Dermatological examination revealed confluent, erythematous plaques with annular scales on the bilateral lower legs, buttocks and lumbar region with no interdigital scaling. The scaly erythema persisted despite antifungal treatment with terbinafine (125 mg/day) and bifonazole cream. Antifungal testing revealed a MIC for terbinafine of 32 µg/mL, while the strain was susceptible to itraconazole and ravuconazole. The treatment then moved to itraconazole (100 mg) and two weeks after switching antifungal agents, the skin lesions resolved [34].

In 2020, Nenoff et al. described twenty-nine cases of dermatophytosis due to *T. mentagrophytes* type VIII diagnosed in Germany (including large cities and rural areas). Patients presented chronic dermatophytosis due to *T. mentagrophytes* type VIII. These cases were diagnosed from 2016–2020 and the patients mainly came from abroad (India, Bahrain, Saudi Arabia, Iraq, Pakistan, etc.). Clinical presentation was mainly tinea corporis. A total of 13/29 (45%) isolated *T. mentagrophytes* were resistant to terbinafine. All the patients were recalcitrant to long-term treatment with terbinafine. All the cases seemed to respond to topical antifungal therapy. However, due to the poor availability of topical drugs in Germany, the author recommend the use of itraconazole 200 mg/day for four to eight weeks [26].

A recent series of extended dermatophytosis due to *T. mentagrophytes* type VIII in an Iranian family (including four people from 5 months to 31 years of age) was described. The isolated *T. mentagrophytes* strain was multidrug resistant with high MICs for terbinafine (≥8 µg/mL), but also for fluconazole (16 µg/mL) and itraconazole (≥4 µg/mL). The extensive lesions were due to the use of corticoid based creams and to intrafamilial transmission. The lesions described for patient 1 (31 year old female) were highly extended itchy, crusted skin lesions that initially emerged in her groin and spread upward over the back, left breast, arms, face and ears. The treatment of this patient with oral terbinafine and topical sertaconazole was not successful. A pulse therapy with itraconazole (200 mg twice daily for one week each month) associated with antifungal shampoo and sertaconazole cream resulted in poor improvement of the lesions. The second patient was a 4 year-old boy (son of patient 1) presenting tinea pedis. Fluconazole was prescribed without any success. A pulse therapy with itraconazole (100 mg twice a day for one week each month) was also ineffective. The third patient was a 30 year old man (brother of patient 1) with a generalized dermatophytosis initially beginning in the groin. As for the other members of the family, oral terbinafine and pulsed itraconazole therapy failed as the lesions reappeared after discontinuing the last drug. Patient 4 was a 64 year old woman (mother of patient 1) with dermatophytosis beginning in her groin. Fluconazole and terbinafine therapies were not successful. Pulsed itraconazole therapy was also initiated but a scaly lesion emerged on the groin and face after stopping the treatment. Finally, in patient 2 the use of two successive itraconazole pulse therapy regimes resulted in complete resolution of the symptoms. In patient 3, the treatment was continued with itraconazole (100 mg/day) for four weeks without recurrence after the end of the treatment. In patients 1 and 4, treatment with voriconazole (200 mg/day) resulted in a complete cure [38].

All these above described cases indicate that certain clinical characteristics are useful indicators of drug resistance, as suggested by Shen et al. 2021. The most important being extended dermatophytosis with multiple relapses following different treatment regimens. The lesions are often large with severe itching. They can be highly or minimally inflammatory. Tinea corporis together with tinea cruris are often described, sometimes with secondary tinea faciei or tinea capitis [48].

### 2.8. Summary of Alternative Therapies for Terbinafine Resistant Strains

Alternative therapies are described with regard to clinical cases in the above sections and are summarized in Table 2 and Table 3. All these described clinical cases demonstrate that the use of prolonged therapy based on other antifungals (azoles for example) has to be considered to treat terbinafine resistant dermatophyte strains giving rise to extended tinea corporis. In their 2021 publication, Gawaz et al. advise increasing the dose of terbinafine to 250 mg/day when resistance is suspected in extended tinea corporis. Another alternative given by the authors is to switch to itraconazole as continuous therapy (200 mg/day) or as a pulse therapy. An increase of the dose of itraconazole to 300–400 mg/day should be considered in cases of azole resistance associated with terbinafine resistance. The use of SUBA-itraconazole (50 mg twice a day) is also an alternative as this new formulation (embedded in a polymer) increases the bioavailability of the drug. Authors also suggest coupling a topical therapy with systemic therapy for better efficacy. Ciclopiroxolamine, amorolfine, miconazole or sertaconazole can be used for this. The duration of the therapy should be between 8–12 weeks and up to one year in cases of multidrug-resistant dermatophytoses [49]. Even the treatment of dermatophytoses caused by terbinafine resistant strains is longer, the use of azoles, especially itraconazole often in combination with a topical treatment, usually permit to have a total recovery of the lesions. However, some cases of relapses at the end of the treatment have been reported with itraconazole. A switch to voriconazole has been efficient in these cases. The use of itraconazole in successive pulsed therapies was also more effective in some cases. To our knowledge, no reports are talking about the alternative therapies when a resistance to azoles and to allyllamines are observed.

In 2021, Shen et al. cited a photodynamic therapy as a treatment option for terbinafine resistant dermatophytes. This photodynamic therapy is a combination of a photosensitizer, light and oxygen to create photoactivated reactive oxygen species that can act against the growth of several microorganisms, including dermatophytes. The team demonstrated that photodynamic therapy treatment in vitro inhibits all dermatophyte isolates, independent of the MIC of terbinafine and the presence of several substitutions in SE. However further in vivo investigations should be conducted to warrant the success of this method for the treatment of dermatophytoses due to resistant strains.

## 3. Focus on Resistance to Azoles in Dermatophytes

Resistance to azoles together with terbinafine resistance has been mentioned above for some cases. We will discuss this in greater detail in the following section.

Resistance to azoles has been reported infrequently until now, but some cases have been described, often together with allylamine resistance. While terbinafine resistance seems to mainly result from action on the squalene epoxidase enzyme implicated in ergosterol synthesis, resistance to azoles can be linked with overexpression of genes encoding ABC transporters, giving rise to multidrug efflux outside the cell. This phenomenon was also mentioned for terbinafine resistance as one alternative to SE point mutation mechanisms [43]. In 2006, Fachin et al. described the probable implication of TruMDR2 in azole/allylamine resistance mechanisms. Their analysis showed increased transcription of TruMDR2 after exposure to antifungals including fluconazole, itraconazole, ketoconazole and tioconazole [43]. Additionally, in 2006, the team observed an increase in the expression level of TruMDR1 when *T. rubrum* was exposed to ketoconazole, fluconazole and itraconazole, suggesting the participation of this gene in drug efflux of azoles in this dermatophyte [50]. In 2019, Monod et al. described in *T. rubrum* that azole resistance is mediated by the ABC transporter TruMDR3. They observed that TruMDR3 could confer azole resistance if overexpressed. They suggest that other ABC transporters could also be implicated in the azole resistance phenomenon, such as TruMDR2 [51]. This supposition was reinforced in 2021, when the same Swiss team performed a deletion of TruMDR2 in *T. rubrum* and observed that TruMDR2 suppression abolished the resistance to itraconazole [52]. In parallel, the same year, the team also showed that major facilitator super family (MFS1), a pleiotropic transporter, also seems to be implicated in azole resistance as the suppression of MFS1 in *T. benhamiae* increases sensitivity to fluconazole and miconazole, while no effect was seen for chloramphenicol [53]. In 2016, Martins et al. demonstrated that ABC transporters act synergistically in dermatophytes and may compensate for one another when challenged with antifungal drugs [54]. In 2018, Kano et al. described expression levels two to four times higher for PDR1, MDR1, MDR2 and MDR4 in a *M. canis* terbinafine-resistant strain [55].

As described in the papers in the first part of the review, the mutations leading to substitutions of amino acids in SE are considered to be the first resistance mechanism in allylamines but not the only one, as cross resistance to azoles and allylamines seems to be linked with double amino acid substitution. Among the most described substitutions, the double substitution F397L/A448T was found. Higher MIC values for fluconazole were observed in these double mutants compared to the single mutant F397L. Kong et al. concluded that A448T alone does not lead to higher resistance to azoles but that the combination of the double substitution F397L/A448T in SE may have a notable impact on triazole MICs [40]. Burmester et al. also suggested that the A448T substitution combined with the F397L substitution in SE is associated with higher MICs for fluconazole and voriconazole, but double mutants Q408L/A448L do not follow this trend [18]. Similar results were found by Ebert et al. This team showed that, among 300 clinical isolates, 10% of *T. mentagrophytes* type VIII presented reduced susceptibility to itraconazole and voriconazole, often associated with the double amino acid substitution in SE F397L/A448T. They described triazoles resistance that was more frequent in terbinafine susceptible isolates, while terbinafine resistance was more frequent in triazoles susceptible isolates. The amino acid substitution A448T probably has an impact on ergosterol synthesis inducing conformational changes and may give rise to reduced susceptibility to azoles [16].

In 2019, Singh et al. defined a unique multidrug resistant clonal *Trichophyton* population distinct from the *T. mentagrophytes/interdigitale* complex that was causing an alarming dermatophytosis outbreak in India. Among this population, resistance to terbinafine was described but this was not the only drug resistance as high MIC values for fluconazole were also reported (32–64 µg/mL) showing the gravity of epidemics in India and the lack of therapeutic alternatives [15].

In 2020, Gaurav et al. described a novel double substitution of the SE gene F397L/Y394N, also associated with high MIC values for itraconazole and fluconazole in *T. mentagrophytes* strains [20].

Nenoff et al., in their retrospective study, described cases of resistance to itraconazole and voriconazole among twenty-nine patients with dermatomycoses due to *T. mentagrophytes* type VIII diagnosed all over Germany from 2016–2020. This resistance was expressed in terbinafine susceptible strains with the amino acid substitution A448T (in three cases) and one strain presented cross resistance to terbinafine, itraconazole and voriconazole with the single amino acid substitution F397L [26].

As mentioned above, in 2021 a multicentre study was conducted and reported by Kong et al. This concerned 135 isolates from India, China, Australia, Germany and the Netherlands. They observed during this study that again the double substitution F397L/A448T was associated with higher MIC values for triazoles in addition to MICs > 16 µg/mL for terbinafine [40]. This reinforces what has been described previously about species sharing double substitutions in SE that seem to be more inclined to develop cross-resistances to azoles and allylamines.

In a paper from 2020, Salehi et al. described the higher efficacy of novel triazoles (luliconazole and lanaconazole) compared to fluconazole and itraconazole. The paper shows that among the sixty dermatophytes tested, the MIC range for new azoles is lower than classically used azoles (0.0005–0.004 µg/mL for luliconazole versus 0.4 to 64 µg/mL for fluconazole) showing that these new azoles can be promising candidates to treat recalcitrant dermatophytosis [56]. Table 4 summarizes the mutations in SE associated with azole resistance.

## 4. Conclusions

This review of the literature shows the importance of performing antifungal susceptibility testing in dermatophytes when there is clinical resistance to standardized treatment. A new Eucast E.Def 11.0 method has been published to help to determine the MICs of antifungals in dermatophytes [57]. It is important to consider this method for MIC determination in routine laboratory testing, especially in recalcitrant cases of onychomycosis but also when an extended tinea corporis is present. Dermatologists should be aware of the emergence of resistant strains of dermatophytes; they should ask the patient for indicative information, such as recent travel to India or surrounding areas when extended tinea corporis is present, and communicate this information to the laboratory and the necessity of realizing an antifungal susceptibility test. If the majority of laboratories have not yet proposed MIC determination for dermatophytes, the national reference centre may help in this purpose with Eucast E.Def 11.0, which is now the most appropriate method to be used and advised for MIC determination in dermatophytes. It is of course mandatory to also include azoles MIC determination, such as itraconazole or voriconazole, which should be considered as alternative therapies when strains are resistant to terbinafine. Considering multiresistant clones, the resistance status for azoles is important. It is important for the national reference centre to follow the emergence of drug resistance by implementing systematic screening in isolated *T. mentagrophytes* strains by phenotypic and genotypic methods to remain aware of the incidence of resistance in different countries. Rapid screening methods can be used for this purpose, as described by Yamada et al. [21]. As the situation in India is alarming, with the incidence of resistance increasing yearly, the situation can rapidly become problematic in other countries, for example in Europe or the USA, if no correct follow up of dermatophytoses is carried out. We here established a list of all amino acid substitutions in SE found in terbinafine resistant dermatophytes. A more frequent follow up of these mutations and actualization of the list of putative substitutions is of importance to assure a good response to treatment in patients concerned by dermatophytoses around the world. The use of Whole Genome Sequencing (WGS) can be an effective tool to screen for SE mutations and find some new mutations associated with terbinafine resistance or other mutated targets implicated in antifungal resistance. We also summarized the main clinical features caused by resistant dermatophytes, we think it is important that dermatologists stay aware about terbinafine resistant dermatophytes and should be able to recognise a case with the typical clinical presentation. This will permit to adapt treatments in cases of suspected terbinafine resistance and rapidly improve the patient management.

## Figures and Tables

**Figure 1 jof-07-00983-f001:**
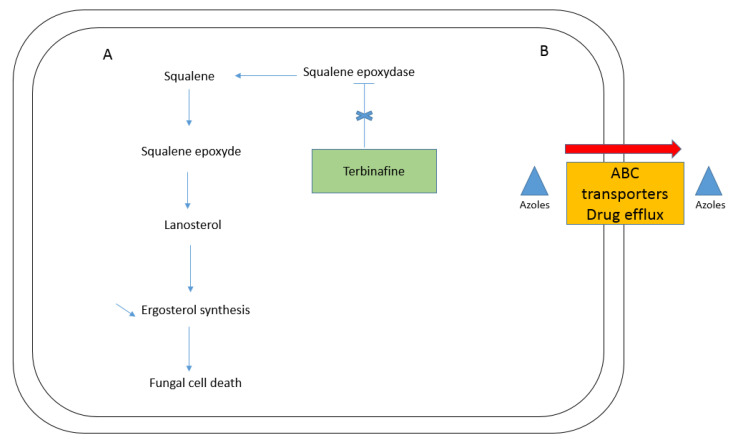
The figure presents the main resistance mechanisms observed in dermatophytes. In (**A**), a reduced scheme of the biosynthesis of ergosterol is represented. In case of mutations in squalene epoxydase gene, the terbinafine cannot inhibit the enzyme anymore, so there is no reduction of ergosterol synthesis and no cell death anymore (no fungicidal effect), giving rise to resistance. In (**B**), the efflux mechanism by ABC transporters is mainly described for azoles resistance in dermatophytes.

**Figure 2 jof-07-00983-f002:**
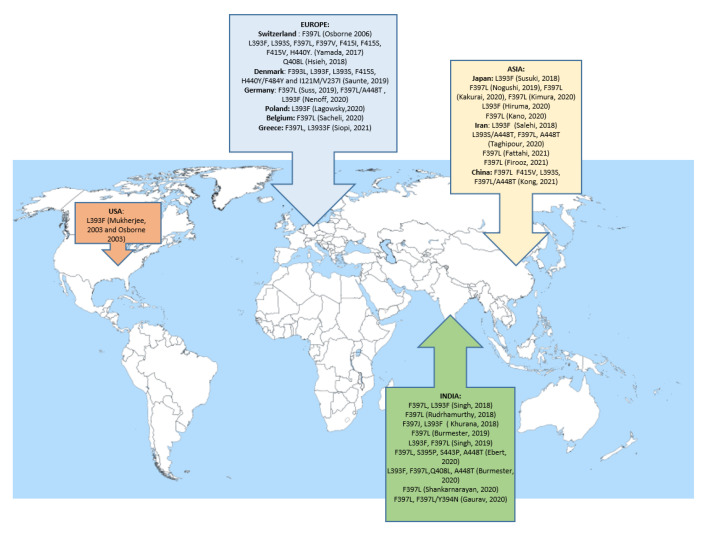
Geographical localization of cases of terbinafine resistance associated with mutations in SE described around the world.

**Table 1 jof-07-00983-t001:** Summary of amino acid substitutions and MICs values observed in terbinafine resistant strains isolated in previous studies. ND = not determined, MIC = minimum inhibitory concentration, SE = squalene epoxidase.

Country of Isolation	Dermatophyte Species	Amino Acid Substitution in SE	Number of Strains with TER Resistance Confirmed by SE Amino Acid Substitution	MIC Terbinafine (µg/mL)	Reference
USA	*T. rubrum*	L393F (6)	6 strains from the same patient	>4	Mukherjee et al., 2003, Osborne et al., 2003 [1]
Switzerland	*T. rubrum*	F397L	1 strain	64	Osborne et al., 2006 [3]
Switzerland	*T. rubrum/T. interdigitale*	In *T. rubrum*: L393F, L393S, F397L, F397I, F397V, F415I, F415S, F415V, and H440Y In *T. interdigitale*: F397L	*T. rubrum*: 16/1664 (1%) *T. interdigitale*: (1/412) (0.2%)	*T. rubrum*: 0.1–12.8 *T. interdigitale*: 3.2	Yamada et al., 2017 [21]
India	*T. interdigitale*	L393F (40%, 8 strains) F397L (60%, 12 strains)	20 strains	4 -> 32	Singh et al., 2018 [11]
India	*T. rubrum/T. nterdigitale*	F397L (6)	*T. rubrum* 2/35 (2.6%) *T. interdigitale*: 4/88 (4.5%)	>2	Rudramurthy et al., 2018 [12]
India	*T. interdigitale*	L393F (3 strains) F397L (10 strains)	13/18 (72.4%) (among complete follow up data and characterised for SE mutation)	0.5–32	Khurana et al., 2018 [13]
India	*T. mentagrophytes*	F397L	1 strain	ND	Burmester et al., 2019 [14]
India	*Indian Trichophyton* spp.	L393F (7 strains) F397L (39 strains)	46 strains/61 (64%)	4–32	Singh et al., 2019 [15]
India	*T. mentagrophytes VIII*	L393S (7), L393F (6), F397L (153) F397L/A448T (27) Q408L/A448T (2) H440T (2) S443P (3) L335F/A448T (1) S396P/A448T (1)	202 strains	0.125/8	Ebert et al., 2020 [16]
India	*T. mentagrophytes*	L393F (1 strain), F397L (6 strains) F397L/A448T (6 strains) Q408L/A448T (1 strain)	14 strains	>=5	Burmester et al., 2020 [18]
India	*T. mentagrophytes*	F397L (15)	15 strains/97 (15.4%)	>=2	Shankarnarayan et al., 2020 [19]
India	*T. mentagrophytes*	F397L (3) F397L/Y394N(1)	4 strains	2–8	Gaurav et al., 2020 [20]
Switzerland (from India)	*T. mentagrophytes*	Q408L	1 strain	>0.2	Hsieh et al., 2019 [17]
Denmark	*T. rubrum/T. interdigitale*	*T. interdigitale*; F397L (1), L393F (1). *T. rubrum*: L393F (1), F397L(6), L393S(2), F415S(1), H440Y/F484Y (1) and I121M/V237I(1)	*T. interdigitale*: 2 strains *T. rubrum*: 12 strains	0.125 -> 8	Saunte et al., 2019 [24]
Germany (from Bahrain)	*T. mentagrophytes VIII*	F397L	1 strain	ND	Suss et al., 2019 [25]
Germany (from India)	*T. mentagrophytes VIII*	F397L	1 strain	ND	Burmester et al., 2019 [14]
Germany (some from India and other surroundings)	*T. mentagrophytes VIII*	F397L (10) L393F (1) F397L/A448T(2)	14 strains/29 (37%)	0.2–16	Nenoff et al., 2020 [26]
Poland	*T. mentagrophytes*	L393F(4)	1 strain in human 3 in foxes	16–32	Lagowski et al., 2020 [27]
Belgium	*T. mentagrophytes*	F397L	1 strain/5	4	Sacheli et al., 2020 [28]
Greece	*T. mentagrophytes type VIII*	F397L (5) L393F (4)	9 strains/24 (37.5%)	0.25–8	Siopi et al., 2021 [29]
Japan	*T. rubrum*	L393F	1 strain	>128	Suzuki et al., 2018 [30]
Iran	*T. rubrum* *T. tonsurans*	L393F(2)	2% of resistant strains in total: *T. rubrum* (1), *T. tonsurans* (1)	>32	Salehi et al., 2018 [31]
Japan	*T. rubrum*	F397L (2)	2 strains from the same patient	>=8	Noguchi et al., 2020 [33]
Japan	*T. interdigitale*	F397L	1 strain	32	Kakurai et al., 2020 [34]
Japan	*T. interdigitale of Indian genotype*	F397L	1 strain	32	Kimura et al., 2020 [35]
Japan	*T. rubrum*	L393F (5)	5 strains/210 (2.4%)	>32	Himura et al., 2020 [36]
Iran	*T.mentagrophytes VIII*	F397L/A448T (4) L393S/A448T(1)	5 strains/45 (11%)	>=32	Taghipour et al., 2020 [37]
Japan (from Nepal/india)	*T. indotineae*	F397L (2)	2 strains	>32	Kano et al., 2020 [41]
Iran	*T. mentagrophytes VIII*	F397L	4 strains from the same family	>=8	Fattahi et al., 2021 [38]
Iran	*T. mentagrophytes*	F397L	1 strain/7 (14.2%)	>8	Firooz et al., 2021 [39]
India, China, Australia, Germany and the Netherland	*T. indotineae*	F397L (25), F397L/A448T (9) F415V (1) L393S (1) H440T (1)	34 strains/64 (53%)	0.125 -> 16	Kong et al., 2021 [40]

**Table 2 jof-07-00983-t002:** Alternative therapies following clinical presentations for terbinafine resistant *T. rubrum*.

Dermatophyte Species	Clinical Presentation	Alternative Therapy	Reference
*T. rubrum*	Tinea unguium	Fosravuconazole (dose undetermined)	Noguchi et al., 2019 [33]
*T. rubrum*	Tinea unguium + tinea palmaris	SUBA-itraconazole 50 mg/day for 1 week then 2 × 50 mg 1×/week + topical ciclopirox	Appelt et al., 2021 [44]
*T. rubrum*	Tinea pedis	Itraconazole 100 mg/day for 3 weeks	Scholser et al., 2018 [23]
*T. rubrum*	Tinea corporis	Itraconazole 100 mg/twice daily	Wingfield Digby et al., 2017 [22]
*T. rubrum*	Tinea corporis, tinea pedis, tinea cruris	Itraconazole 200 mg daily for 3 months	Gu et al., 2020 [42]
*T rubrum*	Tinea corporis, tinea faciei	Itraconazole 100 mg/day for 4 weeks, then 1×/week.	Appelt et al., 2021 [44]
*T rubrum*	Tinea corporis, tinea cruris and tinea glutealis	200 mg every two weeks +bifonazole and cicloprix 1×/day.	Appelt et al., 2021 [44]
*T rubrum*	Tinea corporis	SUBA-itraconazole 2 × 50mg/day for 2 weeks then 1×/week 50 mg for 10 weeks	Appelt et al., 2021 [44]

**Table 3 jof-07-00983-t003:** Alternative therapies following clinical presentations for terbinafine resistant *T. mentagrophytes/indotineae*.

Dermatophyte Species	Clinical Presentation	Alternative Therapy	Reference
*T.mentagrophytes*	Disseminated tinea corporis	Itraconazole 200 mg/day for 2–3 weeks + topical eberconazole	Hsieh et al., 2019 [17]
*T.mentagrophytes*	Tinea, corporis, tinea cruris	Itraconazole + ciclopirox	Burmester et al., 2019 [14]
*T. mentagrophytes VIII*	Extended tinea corporis	Topical miconazole and later ciclopirox	Suss et al., 2019 [25]
*T. mentatgrophytes/interdigitale*	Extensive tinea corporis	Itraconazole 100 mg/day and topical luliconazole	Kimura et al., 2020 [35]
*T. interdigitale*	Extensive tinea corporis	Itraconazole 100 mg/day	Kakurai et al., 2020 [34]
*T. mentagrophytes* VIII	29 cases of tinea corporis	Recommended Itraconazole200 mg/day for 4–8 weeks	Nenoff et al., 2020 [26]
*T. mentagrophytes VIII*	Extended tinea corporis from the groin	Voriconazole 200 mg/day	Fattahi et al., 2021 [38]
*T. mentagrophytes VIII*	Tinea pedis	2 successive itraconazole pulse therapy	Fattahi et al., 2021 [38]
*T. mentagrophytes VIII*	Extended tinea corporis from the groin	Itraconazole 100 mg/day for 4 weeks	Fattahi et al., 2021 [38]
*T. mentagrophytes VIII*	Extended tinea corporis from the groin	Voriconazole 200 mg/day	Fattahi et al., 2021 [38]

**Table 4 jof-07-00983-t004:** Summary of amino acid substitutions in SE and MICs values observed in azoles resistant strains isolated in previous studies. ND = not determined, GM geometric mean, MIC =minimum inhibitory concentration, SE = squalene epoxidase.

Country of Isolation	Dermatophyte Species	Amino acid Substitution in SE	Number of Strains with Azoles Resistance Confirmed by SE Amino-acid Substitution	MIC Azoles or GM Values (µg/mL)	Reference
India, China, Australia, Germany and the Netherland	*T. indotineae*	F397L/A448T A448T F397L	ND	**Itraconazole GM:** F397L/A448T = 0.3 F397L = 0.139 A448T = 0.189 **Fluconazole GM:** F397L/A448T = 32 F397L = 22.32 A448T = 3.48	Kong et al., 2021 [40]
India	*T. mentagrophytes*	F397L/A448T (6) F397L (6) A448T (3) L393F (1) Q408L/A448T(1)	17 strains	MIC fluconazole: >=160	Burmester et al., 2020 [18]
India	*T. mentagrophytes VIII*	F397L/A448T F397L A448T	ND	**Itraconazole GM:** F397L/A448T = 0.26 F397L = 0.10 A448T = 0.19 **Voriconazole GM:** F397L/A448T = 0.26 F397L = 0.05 A448T = 0.15	Ebert et al., 2020 [16]
India	*T. mentagrophytes*	F397L, Y394N (1) F397L (3)	3 strains (fluconazole) 1 strain (fluconazole + itraconazole)	MIC fluconazole: 0.125–128 MIC itraconazole: 2	Gaurav et al., 2020 [20]
Germany (some from India and surroundings)	*T. mentagrophytes VIII*	A448T (3) F397L (1)	Itraconazole (3) Voriconazole (4)	MIC itraconazole: 0.5 MIC voriconazole: 0.25–0.5	Nenoff et al., 2020 [26]

## Data Availability

Not applicable.

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
