# Peer review of "Antifungal Resistance in Dermatophytes: Genetic Considerations, Clinical Presentations and Alternative Therapies"

_jof, 2021, doi:10.3390/jof7110983_

Round 1

Reviewer 1 Report

General comments

The manuscript by Sacheli & Hayette is a well structured and presented concise review on the antifungal resistance in dermatophytes. The paper addresses a timely topic in an balanced manner, effectively conveying main relevant points to the readers. The manuscripts presents an in-depth coverage of the subject also with the help of several referenced tables. In general, I totally agree with presentation and the conclusions of this article, which are mainly in line with the current expert opinions and the current diagnostic guidelines.

My comments/suggestions for improvement are listed below:

Specific comments

Major points

  1. A depiction of the main mechanisms of antifungal resistance in a picture should greatly improve the readability of this manuscript

  2. In addition, a geographical map showing the regions with isolates with antifungal resistance should be a added to a revised manuscripts

  3. The conclusion part should be extented with perspectives of the proposed increase in the antifungal resistance testing for both the routine laboratory diagnostics, therapy of dermatomycoses, and the future research in the area.

Minor points

  1. In the referenced table the reference style is different from the refence format in the manuscript body. Please adjust.

Author Response

Major points

  1. A depiction of the main mechanisms of antifungal resistance in a picture should greatly improve the readability of this manuscript

Answer: Indeed, this a good idea, I inserted a figure 1 describing the two main resistance mechanisms discussed in the paper.

  1. In addition, a geographical map showing the regions with isolates with antifungal resistance should be a added to a revised manuscripts

Answer: I inserted a map in figure 2, with different mutations in squalene epoxydase found around the world and discussed in the paper. This is a complement of table 1 already describing all mutations found in SE following different countries.

  1. The conclusion part should be extented with perspectives of the proposed increase in the antifungal resistance testing for both the routine laboratory diagnostics, therapy of dermatomycoses, and the future research in the area.

 Answer: I modified the conclusion with some perspectives for future research introducing WGS for example. I think that diagnostics perspectives are already well described with the reference to Eucast E. Def.11.0 and the mention of the availabilty of rapid screening tests (with the reference to apply the method, Yamada et al, 2017). For the therapy of dermatomycoses, I added a conclusion in point 2.8 talking about alternative therapies, so I will not repeat it in the main conclusion as it will be redundant.

Minor points

  1. In the referenced table the reference style is different from the refence format in the manuscript body. Please adjust.

Answer: This was adjusted accordingly.

Reviewer 2 Report

The authors present the current state of knowledge on the resistance of dermatophytes to standard antifungal treatments.

In general, this review is well written

Points of criticism :

1a) Paragraphs 2.2 and 2.3 list numerous articles end to end with a short summary of their content. However, a general conclusion is needed for each of these paragraphs

2) The same is true for section 2.7.1. Table 2 lists alternative therapies in case of resistance of Trichophyton rubrum to terbinafine. However, it should be indicated whether these alternative therapies have been successful, as far as this is known.

Specific comment :

Lines 32-33, page 19 : « we find the double substitution…. » A reference is needed and the past must be used. Have the authors worked on this topic?

Author Response

1a) Paragraphs 2.2 and 2.3 list numerous articles end to end with a short summary of their content. However, a general conclusion is needed for each of these paragraphs

 Answer: Ok, I agree, I inserted a conclusion at the end of each cited paragraphs.

2) The same is true for section 2.7.1. Table 2 lists alternative therapies in case of resistance of Trichophyton rubrum to terbinafine. However, it should be indicated whether these alternative therapies have been successful, as far as this is known.

 Answer:  Ok, I agree, I inserted a conclusion at the end of the paragraph.

Specific comment :

Lines 32-33, page 19 : « we find the double substitution…. » A reference is needed and the past must be used. Have the authors worked on this topic?

Answer: This a language mistake and the sentence has been modified as” Among the most-described substitutions, the double substitution F397L/A448T was found.”

Reviewer 3 Report

The manuscript #jof-1460912, entitled “Antifungal resistance in dermatophytes: Genetic considerations, clinical presentations and alternative therapies” by Sacheli and Hayette presents a comprehensive review on (in line with the title) antifungal resistance of dermatophytes. The topic is of a great importance as opportunistic fungal infections (including dermatophytes) still display high mortality rate and the treatment options remain limited. Another issue is the mentioned in the title antifungal resistance. The manuscript is logically planned and reviews multiple reports. The authors concentrated mostly on Trichophyton spp. and the resistance towards terbinafine, inhibitor of squalene epoxidase. Additionally, the authors concentrated on introducing already described point mutations in gene/protein of squalene epoxidase reported throughtout the world, which result in terbinafine resistance. The quality of presentation and overall scientific soundness are on a high level. I, personally, think that such review would be a great, comprehensive "set text" for any mycologist who analyze terbinafine resistance and squalene epoxidase gene point mutantions using clinical isolates. Additionally, such review may be helpful for studies using different fungal species (such as pathogenic yeast), in order to discuss acquired point mutations with the ones collected in this ms for Trichophyton. However, I have only one minor, editorial issue - please pay attention to italicize all fungal names throught the ms. 

Author Response

However, I have only one minor, editorial issue - please pay attention to italicize all fungal names throught the ms. 

Answer: Ok I checked all the manuscript accordingly and italicized if necessary.